# Modulation of Transcription Profile Induced by Antiproliferative Thiosemicarbazone Metal Complexes in U937 Cancer Cells

**DOI:** 10.3390/pharmaceutics15051325

**Published:** 2023-04-24

**Authors:** Serena Montalbano, Franco Bisceglie, Giorgio Pelosi, Mirca Lazzaretti, Annamaria Buschini

**Affiliations:** 1Department of Chemistry, Life Sciences and Environmental Sustainability, University of Parma, Parco Area delle Scienze 11/A, 43124 Parma, Italy; 2COMT (Interdepartmental Centre for Molecular and Translational Oncology), University of Parma, Parco Area delle Scienze 11/A, 43124 Parma, Italy

**Keywords:** thiosemicarbazone metal complexes, molecular action mechanism, expression profile study, ribonucleotide reductase, DNA damage response pathway

## Abstract

Since the discovery of cisplatin, the search for metal-based compounds with therapeutic potential has been a challenge for the scientific community. In this landscape, thiosemicarbazones and their metal derivatives represent a good starting point for the development of anticancer agents with high selectivity and low toxicity. Here, we focused on the action mechanism of three metal thiosemicarbazones [Ni(tcitr)_2_], [Pt(tcitr)_2_], and [Cu(tcitr)_2_], derived from citronellal. The complexes were already synthesized, characterized, and screened for their antiproliferative activity against different cancer cells and for genotoxic/mutagenic potential. In this work, we deepened the understanding of their molecular action mechanism using an in vitro model of a leukemia cell line (U937) and an approach of transcriptional expression profile analysis. U937 cells showed a significant sensitivity to the tested molecules. To better understand DNA damage induced by our complexes, the modulation of a panel of genes involved in the DNA damage response pathway was evaluated. We analyzed whether our compounds affected cell cycle progression to determine a possible correlation between proliferation inhibition and cell cycle arrest. Our results demonstrate that metal complexes target different cellular processes and could be promising candidates in the design of antiproliferative thiosemicarbazones, although their overall molecular mechanism is still to be understood.

## 1. Introduction

Cisplatin was the first metal-based chemotherapeutic drug approved by Food and Drug Administration (FDA) in 1978 [1,2,3] and even today it is administered alone or in combination with radiotherapy or other antineoplastic agents for the treatment of several cancers, such as testicular [4], ovarian [5], lung [6], esophageal [7], stomach [8], prostate [9], cervical [10], colorectal cancers [11], head–neck carcinoma [12], non-Hodgkin lymphoma [13], multiple myeloma [14], melanoma [15] and mesothelioma [16].

The first target of cisplatin cytotoxic activity is represented by genomic (gDNA) or mitochondrial DNA (mtDNA) to create DNA lesions, block DNA replication, mRNA and protein production, and activate several transduction pathways leading to cellular death [17,18]. The literature reports multiple action mechanisms but none of them can explain the actual complete mechanism. In addition to DNA damage, cisplatin induces the production of reactive oxygen species (ROS), such as hydroxyl radicals and superoxide, that cause oxidative stress, lipid peroxidation, depletion of sulfhydryl groups, and changes in different signal transduction pathways. The overcoming of toxic side effects, drug resistance, and relapsing of cisplatin is the major challenge of next-generation anticancer drugs [19,20].

In this landscape, the need to develop novel anticancer agents with strong and selective effects and low toxicity represents an important focus of research because, according to estimates from the World Health Organization (WHO) [21], cancer is currently the second leading cause of death worldwide.

Thiosemicarbazones (TSCs) and their corresponding metal complexes occupy a prominent place in medicinal chemistry and are well-known for their pharmacological activities [22,23,24,25]. The mechanisms by which TSC derivatives exert their antiproliferative effects against cancer cell lines have been linked to various biological activities including ribonucleotide reductase [26,27,28] or topoisomerase II [29,30] inhibition, ROS production [31,32,33] and mitochondria homeostasis alteration [34,35,36].

In addition, TSCs can induce modulation of cellular signaling pathways, cause cell cycle arrest, and affect cell proliferation and death. In this context, cell cycle arrest is one of the main mechanisms that are involved in the development of anticancer compounds. The literature data highlighted that antiproliferative TSCs could affect cell cycle progression: some of them can induce G1/S [37,38] or G2/M [39,40] cell cycle block, while other compounds do not cause cell cycle arrest.

Furthermore, TSCs and their metal complexes affect p53 and p21 protein expression, involved in cell cycle progression and apoptosis induction, increasing the expression of cdc2, and regulating the G2/M phase transition [41]. For example, TSCs derived from the di-2-pyridine ketone and quinoline scaffolds were reported to alter the expression of cyclins, cyclin-dependent kinases (CDKs) p53, and p21CIP/WAF1 [42,43]. These TSCs chelate intracellular iron, disrupting the iron metabolism, affecting the cell cycle progression, and leading to the activation of apoptosis signaling pathways [44]. On the contrary, other TSCs cause a marked decrease in the expression of cyclins [45].

Based on the aforementioned scientific findings, and in continuation of our recent efforts for the development of antiproliferative molecules, in this works we deepened the comprehension of the action mechanisms of a series of TSCs metal complexes derived from the natural aldehyde citronellal and obtained after complexation with nickel [46,47], platinum and copper [48,49]. First, in order to identify a correlation between the well-known antiproliferative activity and a possible cell cycle arrest induced by the tested TSCs, we performed a flow cytometric cell cycle analysis. In addition, we investigated the transcription modulation of the subunits of the enzyme ribonucleotide reductase (RR) as a possible target of the metal complexes because many thiosemicarbazones target RR, interfering with the essential di-iron tyrosyl radical center of its small subunit. Subsequently, since DNA damage induced by metal complexes treatment [46,47,49] could be related to mRNA levels of several proteins regulating DNA damage response and genome integrity, we focused on the DNA damage sensors, Chk1 and Chk2, that participate in G2/M checkpoint control through the ataxia telangiectasia mutated (ATM)/ATM RAD3 related (ATR) pathway. Finally, to determine the correlation between proliferation inhibition and cell cycle blockage, we analyzed the expression levels of cyclin A1, which are known to participate in the initiation of mitosis in human cancer cells, and of cyclin B, a key component involved in G2 to M phase transition.

## 2. Materials and Methods

### 2.1. Synthesis and Chemical Characterization of Nickel [Ni(tcitr)_2_], Platinum [Pt(tcitr)_2_] and Copper [Cu(tcitr)_2_] Complexes

[Ni(tcitr)_2_], [Pt(tcitr)_2_], and [Cu(tcitr)_2_] were synthesized and characterized following the detailed protocol reported in our previous works [46,49,50]. Figure 1 shows the schematic representation of the synthesized metal complexes.

### 2.2. Cell Line and Culture Condition

U937 (ATCC, CRL-1593.2) cells were obtained from the American Tissue Culture Collection (Rockville, MD) and were used for our in vitro studies. Cell culture conditions have been previously reported [46,47,48,49,51,52,53]: briefly, cells were grown in RPMI medium added with 100 U/mL penicillin, 100 µg/mL streptomycin, 2 mM L-glutamine and 10% (*v*/*v*) fetal bovine serum, in a humidified CO_2_ (5%) incubator at 37 °C.

### 2.3. Cell Cycle Analysis on U937 Cell Line by Flow Cytometry

In agreement with several literature experimental studies [37,54,55], to determine distribution of U937 cells in different phases of the cell cycle, DNA content of nuclei was evaluated by propidium iodide staining using flow cytometry. U937 cells were seeded at a concentration of 5 × 10^5^ cell/mL in 1 mL of RPMI complete medium and were incubated at 37 °C in a humidified 5% CO_2_ incubator. Cells were treated with [Pt(tcitr)_2_] and [Cu(tcitr)_2_] at the concentration that reduces U937 cell proliferation to 50% (GI_50_) and incubated for 4 and 24 h. The GI_50_ values were calculated to be 7.0 ± 0.2 µM for [Pt(tcitr)_2_] and 33.0 ± 1.2 µM for [Cu(tcitr)_2_], in good agreement with previous studies [49]. After treatment, cells were collected, washed twice in PBS w/o Ca^2+^ and Mg^2+^, and fixed with 1 mL of 70% ethanol at 4 °C for 2 h. After fixation, cells were washed once in PBS. Cellular pellets were resuspended in 0.5 mL PBS added with 2.5 μL of 1 mg/mL propidium iodide and 2.0 µL of 1 mg/mL RNAse (Sigma-Aldrich, Saint Louis, MO, USA) and were incubated at 37 °C in a water bath for 30 min. For each sample, 12,000 events were analyzed using NovoCyte TM flow cytometry (Agilent, Santa Clara, CA, USA).

### 2.4. mRNA Expression Studies on U937 Cell Line by Real Time qPCR

2 × 10^6^ cells were seeded in flasks with RPMI complete medium and after 24 h were treated with the GI_50_ values of [Ni(tcitr)_2_], [Pt(tcitr)_2_], and [Cu(tcitr)_2_] (10.0 µM, 7.0 µM, and 33.0 µM, respectively) [46,49].

After treatment (1–4–24 h), total RNA was extracted using GeneJET RNA Purification Kit (Thermo Fisher Scientific, Waltham, MA, USA) according to manufacturer’s protocol. RNA concentration and purity were investigated with a Nanodrop 2000 spectrophotometer (Thermo Scientific, Waltham, MA USA). An amount of 1 µg of total RNA was reverse transcribed using QuantiTect^®^ Reverse Transcription Kit (Qiagen, Hilden, Germany) following the manufacturer’s protocol.

The cDNA was used as templates of Real-Time qPCR, using the QuantiNova™ SYBR^®^ Green PCR Kit (Qiagen, Hilden, Germany) and StepOnePlus Real-TimePCR System (Thermo Scientific, Waltham, MA, USA). Three technical replicates were used to validate the amplification specificity. Primer sequences are listed in Table 1 and were obtained from Eurofins Genomics (Rome, Italy).

To analyze the results, target gene expression was normalized to the expression of glyceraldehyde 3-phosphate dehydrogenase gene (GAPDH), selected as internal control after assessing that there were no changes in expression in untreated versus treated cells. The comparative Ct method was used for relative mRNA quantification. Amplification conditions were 95 °C for 2′ to PCR initial heat inactivation, followed by 40 cycles 95 °C for 5″ and 60 °C for 10″.

## 3. Results

### 3.1. Cell Cycle Analysis

To determine the correlation between proliferation inhibition and cell cycle arrest, we analyzed whether the citronellal derivatives affect cell cycle progression. In previous studies, we observed that [Ni(tcitr)_2_] was able to disrupt the cell cycle progression in U937 cells inducing a G2/M cell cycle arrest [46].

Effects of the platinum and copper complexes on the cell cycle were assessed by determining the cell cycle phase distribution of U937 cells after treatment with GI_50_ concentrations of [Pt(tcitr)_2_] (7.0 ± 0.2 µM) and [Cu(tcitr)_2_] (33.0 ± 1.2 µM) for 4 and 24 h. The evaluation was performed by measuring the DNA content of the cells by propidium iodide staining through flow cytometry. After a 4 h treatment with [Pt(tcitr)_2_] and [Cu(tcitr)_2_] no significant differences were found between treated and control cells (Figure 2A). In contrast, after 24 h treatment, [Pt(tcitr)_2_] caused an increase in the S population from 40% to 78% compared to control cells, whereas G1 phase cells decreased from 55% to 21% compared to control cells (Figure 2B). These data suggested that [Pt(tcitr)_2_] could inhibit cell cycle progression inducing a block in the S phase.

On the other hand, [Cu(tcitr)_2_] did not induce cell cycle alterations upon 24 h treatment, in comparison with untreated cells (Figure 2B).

### 3.2. Analysis of mRNA Expression Upon Metal Complexes Treatment by qRT-PCR

To analyze the effect of metal complexes on mRNA expression, U937 cells were treated for 1–4–24 h with GI_50_ values obtained after 24 h of treatment ([Ni(tcitr)_2_] = 10.0 µM, [Pt(tcitr)_2_] = 7.0 µM, [Cu(tcitr)_2_] = 33.0 µM) [46,47,48,49,51,52,53].

To calculate relative changes in gene expression, the 2^−ΔΔCT^ method was performed. qRT-PCR data are presented as fold change in target gene expression in treated cells normalized to the internal control gene (GAPDH) and relative to the DMSO negative control. In the present study, fold change values >2 indicates an upregulation of the target gene, while fold change values <0.5 indicates a downregulation of the target gene. Basal expression was defined by a fold change ranging from 0.5 to 1.9 [56,57].

#### 3.2.1. mRNA Expression Profile of Ribonucleotide Reductase Subunits

First, we investigated the relationship between DNA damage, cell growth inhibition, and ribonucleotide reductase enzyme expression levels to determine if the treatment with metal complexes could involve the transcription of genes coding for RR subunits. Some anticancer TSCs act by inhibiting ribonucleotide reductase, a key enzyme involved in the rate-limiting step of DNA synthesis and responsible for the conversion of ribonucleoside diphosphates to deoxyribonucleotide diphosphates. Furthermore, RR is involved in DNA repair after genotoxic stimuli [58]. RR protein levels are highly expressed in tumor cells rendering this iron-dependent enzyme an excellent target for cancer chemotherapy [59].

Only treatment with [Ni(tcitr)_2_] showed an interesting modulation of the expression of RR subunits: we did not observe alterations in RRM1 expression after the treatment with the nickel complex, while RRM2 showed a very important upregulation at all the exposure times (Figure 3) (Appendix A). On the contrary, [Pt(tcitr)_2_] and [Cu(tcitr)_2_] did not affect the expression of both the subunits of the RR enzyme (Figure 3) (Appendix A).

In previous works, we observed that the metal complexes caused important DNA damage that could be recognized by several DNA damage response pathways [46,49]. After DNA damage induced by genotoxic stress, p53 induces cell-cycle arrest, expression, and nuclear accumulation of p53R2, an additional small subunit, that interacts with RRM1 and has been identified as a transcriptional target of p53. The mechanism by which p53R2 activity is induced by p53 may not be rapid enough to supply dNTPs to prompt DNA repair, which can be completed within a few hours after DNA damage [60,61]. In this context, to understand if the cellular response to DNA damage induced by the metal complexes could involve the p53 pathway, we analyzed the expression of the p53R2 subunit of the RR: in this case, [Cu(tcitr)_2_] did not disrupt the basal level of expression of p53R2; while [Ni(tcitr)_2_] and [Pt(tcitr)_2_] caused a mild downregulation of gene expression at 1 and 4 h treatments (Figure 3) (Appendix A).

#### 3.2.2. mRNA Expression Profile of DNA Damage Response Pathway

We examined the changes in mRNA expression of two distinct kinase signaling cascades, the ATM-Chk2 and ATR-Chk1 pathways that are activated by DSBs (DNA double-strand breaks) and SSBs (DNA single-strand breaks), respectively.

The early DNA damage induced by [Ni(tcitr)_2_] upon a 1 h treatment [46,47] was able to activate Chk2 transcription, as shown by our results: the treatment with [Ni(tcitr)_2_] in U937 cells caused ATM downregulation after 1 h followed by the restoring of the basal expression level after 4 and 24 h. On the contrary, we observed an early and strong upregulation of Chk2 (Figure 4) (Appendix A).

[Pt(tcitr)_2_] did not modify ATM and ATR-Chk1 expression, indicating that the de novo transcription of these kinases was not activated (Figure 4) (Appendix A). At the same time, the Chk2 upregulation observed after 24 h treatment could indicate that the DNA damage induced by [Pt(tcitr)_2_] could be recognized only later by the DDR pathway (Figure 4) (Appendix A).

Regarding the gene expression modulation induced by the [Cu(tcitr)_2_] treatment, the ATM-Chk2 pathway and ATR expression were severely downregulated (Appendix A). The most important result was the Chk1 modulation induced by the copper complex in U937 cells: despite the downregulation observed after 1 and 4 h treatment, the gene has undergone a strong upregulation over 24 h (Figure 4) (Appendix A).

#### 3.2.3. mRNA Expression Profile of Cyclins

To determine the correlation between the proliferation inhibition and the possible cell cycle block, we analyzed the expression levels of cyclin A1 and B.

The nickel complex preserved the basal gene expression of cyclin A1, while a decrease in cyclin A expression was observed after 4 and 24 h treatment with [Cu(tcitr)_2_] (Figure 5) (Appendix A). An interesting upregulation of the cyclin A gene has been observed after 24 h treatment with [Pt(tcitr)_2_] (Appendix A).

Only the treatment with the nickel complex has led to an increase in the mRNA level of cyclin B (Figure 5) (Appendix A).

## 4. Discussion

As a general statement, we can say that the anticancer effects of TSC metal complexes are closely related to their chemical structure, to the nature of the substituents, and to the cancer cell types.

Several thiosemicarbazones inhibit the small subunit of the ribonucleotide reductase. This metalloenzyme is crucial for DNA synthesis as well as for DNA damage repair and is frequently overexpressed in cancer cells making it an attractive target for the treatment. Inhibition of the tyrosyl was radical in the active center of the RRM2/p53R2 subunit was demonstrated for several thiosemicarbazones (including Triapine and Dp44mT). Furthermore, previous studies in *Saccharomyces cerevisiae*, carried out to clarify the action mechanism of [Ni(tcitr)_2_], have shown that, after the analysis of a collection of deletants, enrichment in the classes of genes coding for components involved in nucleic acids metabolism such as ribonucleotide reductase is observed [57]. In U937 cells, the nickel complex induced a very significant modulation of the different subunits of RNR. Probably, the strong RRM2 upregulation could indicate that the compound could act targeting specifically this subunit of the enzyme. We also analyzed p53R2 expression in order to determine if the cellular response induced by the nickel complex could involve a p53 activity. We did not observe an alteration in the transcription level of the p53R2 subunit and this result highlighted that the [Ni(tcitr)_2_] may act with a p53-independent mechanism, as hypothesized before in our analysis [46,47] and as confirmed by other literature studies [43,44]. Indeed, several novel TSCs disrupt cell cycle progression and induce apoptosis through a p53-independent mechanism [43,44,62].

We focused then on the mechanism of action of the different metal complexes derived from citronellal thiosemicarbazone. Identifying molecular pathways targeted by a compound is of paramount importance for the development of new drugs and the prediction of its mechanism of action has been attempted by using transcriptional expression profiles following drug treatment [63]. In previous studies, [Ni(tcitr)_2_] showed interesting antiproliferative characteristics towards histiocytic lymphoma cell line U937. It induced G2/M cell cycle arrest and apoptosis by downregulation of Bcl-2, mitochondrial membrane potential loss, and caspase activation. [Ni(tcitr)_2_] displayed DNA damaging potential, but it was not due to DNA oxidation. [Ni(tcitr)_2_] did not induce gene mutation or chromosomal damage but altered DNA conformation creating knot-like structures and hairpins [47]. We previously assumed that the DNA damage induced by the nickel complex could be due to a direct interaction between the complex and the DNA backbone and/or histones, giving rise to structural alterations of chromatin, such as heterochromatinization, that could interfere with correct mitosis processes inducing apoptosis [46,47].

To understand if DNA alteration could be related to real DNA damage or to an altered DNA conformation that could produce alkali labile sites, we analyzed the transcription profile of genes involved in DDR, such as ATM, ATR, Chk1, Chk2, cyclin A1, and B. Under genotoxic stress, the activation of ATM and/or ATR, DNA sensors for the initial response to single and double-stranded DNA breaks, is a result of the formation of their monomers or the induction of their transcription. After the treatment with the nickel complex, we did not observe an activation of ATM and/or ATR transcription. These results corroborate our hypothesis that [Ni(tcitr)_2_] does not induce SSB and/or DSB breaks, directly or indirectly. These data are also in agreement with the lack of in vitro clastogenic activity of [Ni(tcitr)_2_] on supercoiled DNA plasmid pBR322, previously reported [47]. We did not observe alterations in the mRNA levels of ATM and ATR, but in contrast, we found a strong upregulation of the DNA damage sensors, Chk1 and Chk2. Probably, the metal complex could activate alternative pathways that trigger the transcription of Chk1 and Chk2. Indeed, the DNA damage and the cellular alterations induced by the nickel complex were able to activate the transcription of Chk2 already upon a 1 h treatment, while the cellular pathway triggered by the platinum complex showed a long-term effect due to the Chk2 upregulation after a 24 h treatment. On the contrary, [Cu(tcitr)_2_] induced a significant upregulation of Chk1, despite did not cause an increase in ATR expression.

[Ni(tcitr)_2_] caused an important upregulation of cyclin B, that together with Cdk1, is involved in the cell cycle progression. In stress conditions, cyclin B could interact with APC (anaphase-promoting complex), which plays a central role in regulating mitosis and the G1 phase of the cell cycle. Furthermore, APC induces the degradation of cyclin B and inhibits cell cycle progression. This upregulation could lead to the cell cycle block and could interfere with the normal transition from the G2 phase to the M phase, as previously reported for [Ni(tcitr)_2_] [46,47]. These results support a strong interference of [Ni(tcitr)_2_] with the correct folding of the chromosome during mitosis leading to apoptosis. Further analysis of the proteins involved in the mitotic checkpoint could be important to better understand the relationship between DNA—[Ni(tcitr)_2_] interactions and cellular toxicity.

We used the same approach to identify the molecular action mechanisms of the copper and the platinum complexes. In previous work, we highlighted that both induced significant DNA damage [49]. We presume that the DNA damage observed was not recognized by the DDR pathway since we did not observe an activation of the transcriptional profile of ATM and ATR, but we noticed an upregulation of Chk1. In addition, [Cu(tcitr)_2_] was not able to induce a cell cycle arrest [48]. Probably, the action mechanism of [Cu(tcitr)_2_] could involve an excessive production of ROS species, and oxidative stress due to ROS is known to cause DNA lesions (SSB and DSB) through the direct interaction of ROS with DNA [64].

In previous work, we demonstrated that also the platinum complex induces strong DNA damage and acts as a promutagen agent, and it is known that most platinum anticancer agents target DNA [65]. We, therefore, studied the relationship between the cell cycle and cell death investigating if the treatment with the platinum complexes could involve an alteration in the progression of the cell cycle. No ATM and ATR transcription were detected, while Chk2 was upregulated after 24 h treatment. Chk2 is directly involved in G1-S cell cycle arrest. Cisplatin usually activates DDR through the involvement of ATM. Presently it is not known how [Pt(tcitr)_2_] is able to modulate the cell cycle progression without modulating ATM transcription.

This work highlights that the biological activity of TSC complexes strongly depends on the metal ion and that metal ions play a key role in the anticancer activity of TSC metal complexes. These compounds, therefore, represent an emerging class of experimental anticancer agents that shows various in vitro antiproliferative activities and act as multitarget agents. As we previously reported, the predominant target is probably DNA, but our data show that the metal complexes are also able to trigger several cellular pathways involved in DNA damage response and in the cell cycle progression.

In conclusion, these new experimental data confirm that transition metal complexes containing the thiosemicarbazone scaffold represent a good starting point for the development of new anticancer agents. Nevertheless, to have a clear view of the action mechanism of our newly synthesized compounds further studies on the expression of proteins involved in the DDR pathway are still required.

## Figures and Tables

**Figure 1 pharmaceutics-15-01325-f001:**
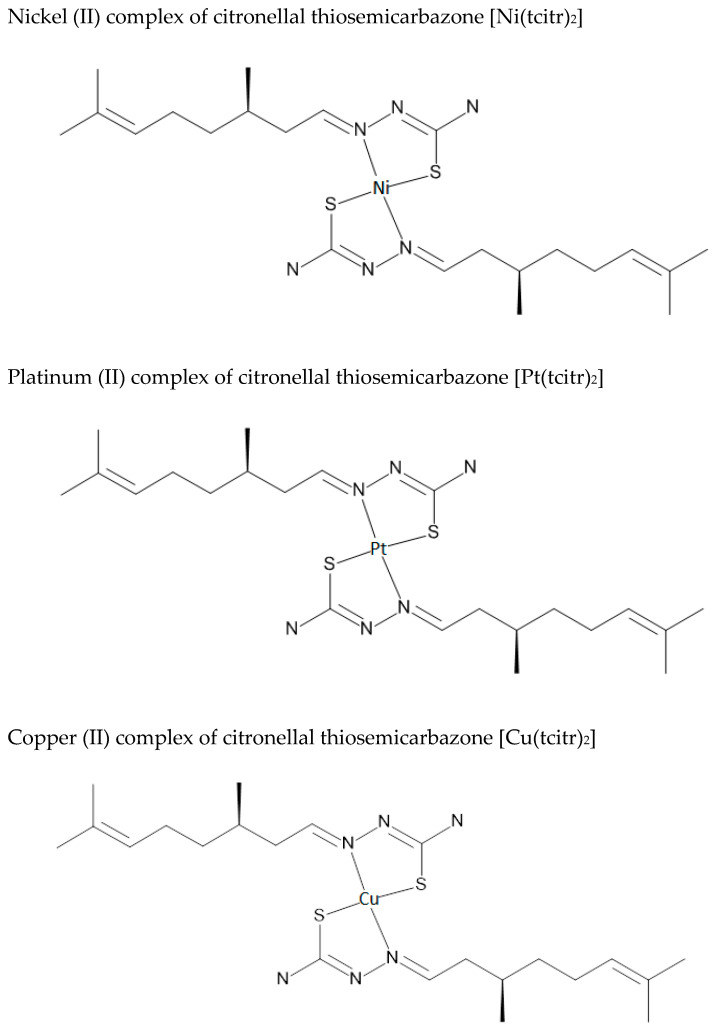
Schematic representation of the synthesized metal complexes [Ni(tcitr)_2_], [Pt(tcitr)_2_], and [Cu(tcitr)_2_].

**Figure 2 pharmaceutics-15-01325-f002:**
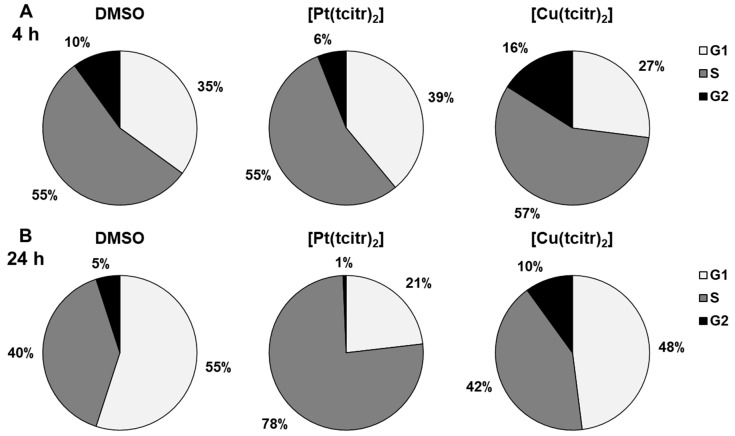
Flow cytometric cell cycle analysis by NovoCyte TM flow cytometry. U937 cells were treated for 4 (**A**) and 24 (**B**) h with the GI_50_ values of [Pt(tcitr)_2_] (**A**) and [Cu(tcitr)_2_]. Data are expressed as percentage of cells in G1, S, and G2 phases of cell cycle. DMSO: negative control; [Pt(tcitr)_2_]: cells treated with [Pt(tcitr)_2_] at 7.0 µM for 4 (**A**) and 24 h (**B**); [Cu(tcitr)_2_]: cells treated with [Cu(tcitr)_2_] at 33.0 µM for 4 (**A**) and 24 h (**B**).

**Figure 3 pharmaceutics-15-01325-f003:**
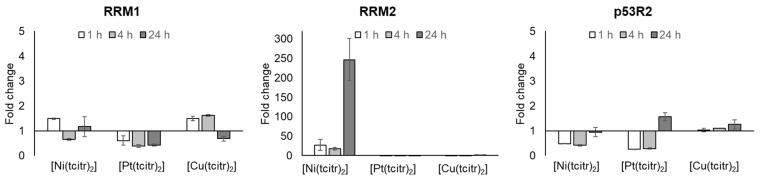
Effects of [Ni(tcitr)_2_], [Pt(tcitr)_2_], and [Cu(tcitr)_2_] on the expression of RRM1, RRM2, and p53R2. U937 cells were seeded (2 × 10^6^ cells/flask) into 25 cm^2^ flasks with complete medium and then treated with GI_50_ value for each metal complex for 1–4–24 h. Total RNA was extracted, quantified, and 1 μg was reverse-transcribed. The complementary DNA (cDNA) was used as a template for qRT-PCR reactions. Data are expressed as fold change ± standard deviation in target gene expression in treated cells normalized to the internal control gene (GAPDH) and relative to the DMSO negative control.

**Figure 4 pharmaceutics-15-01325-f004:**
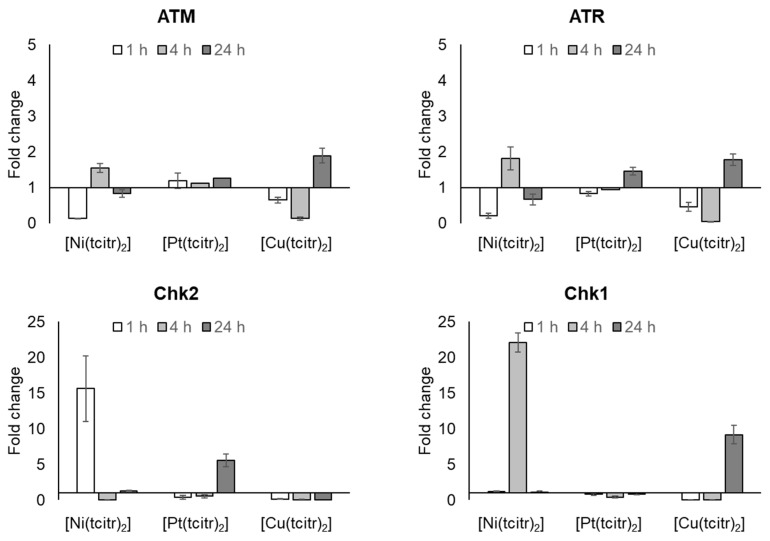
Effects of [Ni(tcitr)_2_], [Pt(tcitr)_2_], and [Cu(tcitr)_2_] on the expression of ATM, ATR; Chk2 and Chk1. U937 cells were seeded (2 × 10^6^ cells/flask) into 25 cm^2^ flasks with complete medium and then treated with GI_50_ value for each metal complex for 1–4–24 h. Total RNA was extracted, quantified, and 1 μg was reverse-transcribed. The complementary DNA (cDNA) was used as a template for qRT-PCR reactions. Data are expressed as fold change ± standard deviation in target gene expression in treated cells normalized to the internal control gene (GAPDH) and relative to the DMSO negative control.

**Figure 5 pharmaceutics-15-01325-f005:**
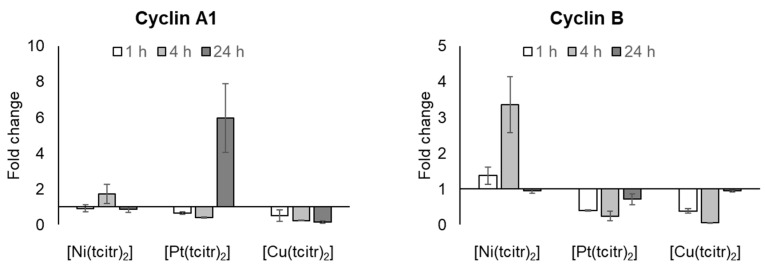
Effects of [Ni(tcitr)_2_], [Pt(tcitr)_2_], and [Cu(tcitr)_2_] on the expression of cyclin A1 and cyclin B. U937 cells were seeded (2 × 10^6^ cells/flask) into 25 cm^2^ flasks with complete medium and then treated with GI_50_ value for each metal complexes for 1–4–24 h. Total RNA was extracted, quantified, and 1 μg was reverse-transcribed. The complementary DNA (cDNA) was used as a template for qRT-PCR reactions. Data are expressed as fold change ± standard deviation in target gene expression in treated cells normalized to the internal control gene (GAPDH) and relative to the DMSO negative control.

**Table 1 pharmaceutics-15-01325-t001:** Primer sequences selected for qPCR.

	Primer Forward (5′-3′)	Primer Reverse (5′-3′)
*GAPDH*	ATGACATCAAGAAGGTGGTG	CATACCAGGAAATGAGCTTG
*RRM1*	AAGAGCAGCGTGCCAGAGAT	ACACATCAAAGACCATCCTGATTAG
*RRM2*	ACCAACTAGCCACACACCATGA	GGACTGTTTAATCCCGCTGT
*p53R2*	CCTTGCGATGGATAGCAGATAGA	GCCAGAATATAGCAGCAAAAGATC
*Cyclin A1*	GTCAGAGAGGGGATGGCAT	CCAGTCCACCAGAATCGTG
*Cyclin B*	CGGGAAGTCACTGGAAACAT	AAACATGGCAGTGACACCAA
*Chk1*	GGTGCCTATGGAGAAGTTCAA	TCTACGGCACGCTTCATATC
*Chk2*	CGGATGTTGAGGCTCACGA	TATGCCCTGGGACTGTGAGG
*ATM*	CAGCAGCTGTTACCTGTTTG	TAGATAGGCCAGCATTGGAT
*ATR*	TGTCTGTACTCTTCACGGCATGTT	AAGAGGTCCACATGTCCGTGTT

## Data Availability

Not applicable.

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
