# Peer review of "Modulation of Transcription Profile Induced by Antiproliferative Thiosemicarbazone Metal Complexes in U937 Cancer Cells"

_pharmaceutics, 2023, doi:10.3390/pharmaceutics15051325_

Round 1
Reviewer 1 Report
To Authors:
Following items below are this reviewer suggestions to address in order for consideration for publication:
The proposed 3 thiosemicarbazones: [Ni(tcitr)2], [Pt(tcitr)2] and [Cu(tcitr)2] should be screened for cellular Cell Titer Glo and or MTT assay in 10-doses for IC50s. Current Flow Cytometry data to be supported by IC50s to confirm its potency in this cell line.
There are quite a few leukemia cells available from ATCC and DSZM, authors should justify for the reason U937 cells line was chosen in your entire study.
Other Leukemic cell line transcriptional expression profile analysis results and compare with chosen U937 cell line which is critical.
Author Response
Reviewer 1
Following items below are this reviewer suggestions to address in order for consideration for publication:
- The proposed 3 thiosemicarbazones: [Ni(tcitr)2], [Pt(tcitr)2] and [Cu(tcitr)2] should be screened for cellular Cell Titer Glo and or MTT assay in 10-doses for IC50s. Current Flow Cytometry data to be supported by IC50s to confirm its potency in this cell line.
As we say in the introduction, we have already published the biological data obtained through MTS assay in U937 cells after 24h treatment with [Ni(tcitr)2], [Pt(tcitr)2] and [Cu(tcitr)2]. The dose-response curves for each thiosemicarbazone have been determined after the evaluation of the cytotoxicity of the metal complexes in a wide range of concentrations. According to the National Cancer Institute approach for the “60 human tumor cell line anticancer drug screen” (NCI60), growth inhibiting concentrations at 50% (GI50) have been calculated. [Buschini A.; Pinelli S.; Pellacani C.; Giordani F.; Ferrari M.B.; Bisceglie F.; Giannetto M.; Pelosi G.; Tarasconi P. Synthesis, characterization and deepening in the comprehension of the biological action mechanisms of a new nickel complex with antiproliferative activity. J Inorg Biochem. 2009;103(5):666-77. doi: 10.1016/j.jinorgbio.2008.12.016] [Bisceglie F.; Orsoni N.; Pioli M.; Bonati B.; Tarasconi P.; Rivetti C.; Amidani D.; Montalbano S.; Buschini A.; Pelosi G. Cytotoxic activity of copper(ii), nickel(ii) and platinum(ii) thiosemicarbazone derivatives: interaction with DNA and the H2A histone peptide. Metallomics. 2019 Oct 16;11(10):1729-1742. doi: 10.1039/c9mt00166b]. Furthermore, as suggested by the Reviewer, we have already planned to investigate the cytotoxicity of our metal complexes with a flow cytometry-based assay.
- There are quite a few leukemia cells available from ATCC and DSZM, authors should justify for the reason U937 cells line was chosen in your entire study.
We used human leukocytes U937 to define a toxicological profile of our compounds on human blood cells. In our previous works, we employed this cell model to investigate the antiproliferative activity and the action mechanism of newly synthesized molecules. Currently, these cells represent in our lab the first in vitro model for a preliminary screening of biological activity of new metal-based compounds. In addition, our previous studies showed the cytotoxicity of the metal complexes against different cancer cell lines (Jurkat, HT29, MCF-7) [Baruffini E, Ruotolo R, Bisceglie F, Montalbano S, Ottonello S, Pelosi G, Buschini A, Lodi T. Mechanistic insights on the mode of action of an antiproliferative thiosemicarbazone-nickel complex revealed by an integrated chemogenomic profiling study. Sci Rep. 2020 Jun 29;10(1):10524. doi: 10.1038/s41598-020-67439-y].
- Other Leukemic cell line transcriptional expression profile analysis results and compare with chosen U937 cell line which is critical.
We agree with the Reviewer about the importance to perform additional transcriptional expression profile analysis on a panel of cancer cells deriving from different districts. In our previous work, we published results of RRM1 and RRM2 expression in Jurkat (acute T cell leukemia) and HT29 (colorectal adenocarninoma) cells, showing that the amount of RNR transcripts and [Ni(tcitr)2] sensitivity [Baruffini E, Ruotolo R, Bisceglie F, Montalbano S, Ottonello S, Pelosi G, Buschini A, Lodi T. Mechanistic insights on the mode of action of an antiproliferative thiosemicarbazone-nickel complex revealed by an integrated chemogenomic profiling study. Sci Rep. 2020 Jun 29;10(1):10524. doi: 10.1038/s41598-020-67439-y].
This work represents a preliminary investigation to deepen our understanding of the possible action mechanisms of our compounds in U937 cells, but the detailed mechanistic effects of the metal complexes will be explored in further studies. At present, we can assume that our compounds could act as multitarget agents and could trigger several cellular pathways.

Reviewer 2 Report
This paper described a decent study of the mechanism of the antiproliferative properties of a family of thiosemicarbazone metal complexes. Though this is not novel, the studies carried out in this current report are scientifically sound and of some value. All in all, I will recommend its publication.
My one and only concern will be whether the inference from this study can be extrapolated for several, other malignant cell lines (other than U937 used in this work). I would appreciate a response from the authors in this regard.
Overall, I recommend its publication.
Author Response
Reviewer 2
This paper described a decent study of the mechanism of the antiproliferative properties of a family of thiosemicarbazone metal complexes. Though this is not novel, the studies carried out in this current report are scientifically sound and of some value. All in all, I will recommend its publication.
- My one and only concern will be whether the inference from this study can be extrapolated for several, other malignant cell lines (other than U937 used in this work). I would appreciate a response from the authors in this regard.
U937 cells were selected in our lab as in vitro model to investigate and deepen our comprehension of the cytotoxicity and the action mechanisms of [Ni(tcitr)2], [Pt(tcitr)2] and [Cu(tcitr)2]. Our previous screening to explore the cytotoxic profile showed that leukemia cells are among the most sensitive cell lines and for this reason we have been planned to carry out this study. We analysed the transcriptional profile of a panel of genes involved in the DDR pathway and the cell cycle regulation that were modulated by our compounds. Although cancer cell lines are essential for in vitro cancer research and drug discovery, each of them shows different mechanisms of drug response that do not allow to standardize and to elucidate the overall complex relationships between molecular and genetic profiles and therapeutic response. In our opinion, further several analyses will be necessary to better understand if the effects induced by our metal complexes could be extrapolated for other cancer cells.

Reviewer 3 Report
The manuscript entitled “Modulation of transcription profile induced by antiproliferative thiosemicarbazone metal complexes in U937 cancer cells” by Serena Montalbano and research team members describes the action mechanism of nickel, platinum, and copper complexes containing thiosemicarbazones, derived from citronellal. These three complexes have already been reported in the literature in synthesis and their characterization including antiproliferative against different cancer cell lines. In the current paper, authors focus on molecular action mechanisms using in vitro models of a leukemia cell line (U937) and an approach of transcriptional expression profile analysis including other bioassays. I have noticed the following major lacunae in the manuscript which need to be addressed before accepting this manuscript for publication in Pharmaceutics:
Comments:
1. In the abstract, the authors need to explain what type of thiosemicarbazone is used in this paper. Most importantly, they should include some antiproliferative results in this part.
2. In the introduction, the authors should include the importance of thiosemicarbazone and their derivatives for biological applications Authors should cite the most relevant papers related to thiosemicarbazone for various biological applications. For example, Journal of Molecular Structure 1250 (2022) 131782.
3. In the Materials and Methods or Results and Discussion sections authors should include the chemical structure of thiosemicarbazone and Ni, Pt, and Cu complexes for better understanding of readers.
4. In the Materials and Methods, Cell line and culture condition, and Cell cycle analysis authors need to include some relevant papers to support these experiments.
5. Authors need to include complexes' activity in detail and compared with previously reported papers.
6. In Figures 2, 3, and 4, the general formula of complexes should be correct. The current formula is not correct. It is very important.
Author Response
Reviewer 3
The manuscript entitled “Modulation of transcription profile induced by antiproliferative thiosemicarbazone metal complexes in U937 cancer cells” by Serena Montalbano and research team members describes the action mechanism of nickel, platinum, and copper complexes containing thiosemicarbazones, derived from citronellal. These three complexes have already been reported in the literature in synthesis and their characterization including antiproliferative against different cancer cell lines. In the current paper, authors focus on molecular action mechanisms using in vitro models of a leukemia cell line (U937) and an approach of transcriptional expression profile analysis including other bioassays. I have noticed the following major lacunae in the manuscript which need to be addressed before accepting this manuscript for publication in Pharmaceutics:
- In the abstract, the authors need to explain what type of thiosemicarbazone is used in this paper. Most importantly, they should include some antiproliferative results in this part.
We thank the Reviewer for the observation. We modified the abstract accordingly.
- In the introduction, the authors should include the importance of thiosemicarbazone and their derivatives for biological applications Authors should cite the most relevant papers related to thiosemicarbazone for various biological applications. For example, Journal of Molecular Structure 1250 (2022) 131782.
In the introduction, following the reviewer’s suggestion, we have added an overall review on the SAR of the biological properties of thiosemicarbazones (the reference suggested by the reviewer unfortunately does not have to do with biological properties) [Pelosi G. Thiosemicarbazone Metal Complexes: From Structure to Activity. The Open Crystallography Journal, 2010, 3, 16-28. doi: 10.2174/1874846501003010016].
- In the Materials and Methods or Results and Discussion sections authors should include the chemical structure of thiosemicarbazone and Ni, Pt, and Cu complexes for better understanding of readers.
As recommended by the Reviewer, in the section of Materials and Methods we added a figure to show the chemical structure of our metal complexes.
- In the Materials and Methods, Cell line and culture condition, and Cell cycle analysis authors need to include some relevant papers to support these experiments.
We thank the Reviewer for having underlined this issue and we added other relevant papers to support our experiments and results.
- Authors need to include complexes' activity in detail and compared with previously reported papers.
As suggested by the Reviewer, we deepened the discussion.
In Figures 2, 3, and 4, the general formula of complexes should be correct. The current formula is not correct. It is very important.
We modified the general formula of complexes in the figures.

Round 2
Reviewer 3 Report
The authors responded satisfactorily to my questions, thus I am happy to suggest the publication of the manuscript in the Pharmaceutics journal.